# Parallelised Diffeomorphic
# Sampling-based Motion Planning

**Tin Lai**[*,1], **Weiming Zhi**[1]**, Tucker Hermans**[2,3] **and Fabio Ramos**[1,3]
[*]Correspondence to `tin.lai@sydney.edu.au` [1]School of Computer Science,
The University of Sydney [2]School of Computing, University of Utah [3]NVIDIA, USA

**Abstract:** We propose Parallelised Diffeomorphic Sampling-based Motion Planning (PDMP). PDMP is a novel parallelised framework that uses bijective and differentiable mappings, or *diffeomorphisms*, to transform sampling distributions of sampling-based motion planners, in a manner akin to normalising flows. Unlike normalising flow models which use invertible neural network structures to represent these diffeomorphisms, we develop them from gradient information of desired costs, and encode desirable behaviour, such as obstacle avoidance. These transformed sampling distributions can then be used for sampling-based motion planning. A particular example is when we wish to imbue the sampling distribution with knowledge of the environment geometry, such that drawn samples are less prone to be in collisions. To this end, we propose to learn a continuous occupancy representation from environment occupancy data, such that gradients of the representation defines a valid diffeomorphism and is amenable to fast parallel evaluation. We use this to "morph" the sampling distribution to draw far fewer collision-prone samples. PDMP is able to leverage gradient information of costs, to inject specifications, in a manner similar to optimisation-based motion planning methods, but relies on drawing from a sampling distribution, retaining the tendency to find more global solutions, thereby bridging the gap between trajectory optimisation and sampling-based planning methods.

**Keywords:** Sampling-based motion planning, diffeomorphism, flows, Sampling distribution, RRT, PRM

## 1  Introduction

This paper addresses the problem of motion planning, and bridges together two motion planning paradigms: *trajectory optimisation approaches* and *sampling-based approaches*. Trajectory optimisation views robot trajectories as solutions of an optimisation problem. The optimisation problem typically incorporates the environment occupancy, along with additionally specified requirements into a cost function. Gradients of the cost function are often assumed to be accessible, allowing for its efficient optimisation. However, trajectory optimisation approaches are known to suffer from local minima, and are generally not *anytime*. On the other hand, sampling-based planners have a complementary set of benefits. Sampling-based planners are probabilistically complete, and are able to quickly find a feasible solution and improve upon it. However, unlike trajectory optimisation approaches, sampling-based planners are unable to utilise gradient information, either from environment occupancy or from user specification.

We propose a novel motion planning framework, Parallelised Diffeomorphic Sampling-based Motion Planning (PDMP), which combines the benefits of both sampling-based and trajectory optimisation methods. PDMP is capable of finding globally optimal solutions, while benefiting from gradient information of cost functions to speed up motion planning. In broad strokes, our method leverages gradient information from specified cost functions, which can be learned from environment data or user specified, to construct differentiable bijections, or *diffeomophisms*. Like normalising flows, PDMP uses diffeomorphisms to shape a simple base sampling distribution into a sampling distribution that is more informative. However, to learn invertible transformations for normalising flows, one typically assumes that samples from a desired target distribution are accessible. This is typically not

5th Conference on Robot Learning (CoRL 2021), London, UK.

the case when learning sampling distributions: generally, we are given information about the occupancy in the environment, as well as designed costs, rather than samples from a "good" sampling distribution. We demonstrate that with relatively mild assumptions, we can obtain diffeomorphisms, from provided cost functions, that allows us to deform the sampling distribution. We provide a specific example of learning a diffeomorphism that conforms to environment occupancy.

The "morphing" of sampling distributions allows sampling-based planners, such as Rapidly-exploring Random Trees (RRTs) and its variants, to more efficiently create connections, speeding-up the planning process. Additionally, the transformation of the sampling distribution can be viewed as a parallel process, while building the expanding trees is an inherently sequential process. We integrate CPU and GPU parallelism: we use GPUs to shape the sampling distribution, which can be processed in parallel, while simultaneously using the CPU to build the expanding tree, which is a sequential process.

Concretely, our contributions are as follows: (i) We propose a method of shaping sampling distributions of sampling-based planners, such that gradient information, for example from environment occupancy gradient or user specified information can be incorporated, allowing for faster motion planning; (ii) We provide an efficient implementation of our method integrated into an RRT motion-planner which leverages the parallel capabilities of GPUs. We demonstrate that the shaping of the sampling distribution can be done efficiently in parallel in a GPU, simultaneously with the sequential tree-building process resulting in no additional time cost.

We empirically evaluate PDMP on challenging planning scenarios, and find that it is able to consistently find solutions faster than existing sampling-based motion planners.

## 2    Related Works

**Sampling-based Motion Planning:** Sampling-based planners are a class of predominant methods to compute motion trajectories for robots. They pose the motion planning problem in a probabilistic setting, where the construction of motion plans are formulated as a graph or tree building procedure. PRM [1] was first proposed to creates a random roadmap of connectivity in the configuration space to avoid the curse of dimensionality. On the other hand, tree-based RRT [2] follows a similar idea but uses tree structures to obtaining solution quicker, which inspire a whole new class of motion planning methods [3, 4, 5, 6, 7]. Since sampling is one of the core component in sampling-based motion planners, there are a lot of methods that tries to improve the sampling distribution. For example, formulating a restricted distribution to improve planning time [8, 9]. There are also method to learn the sampling distribution from experience using neural network methods [10, 11, 12]. However, most learning-based methods learns a skewed distribution based purely from a subset of successful motion plans configurations, which requires a mixture with uniform distribution to maintain the *probabilistic completeness* guarantee [13].

**Gradient-based Cost Optimisation in Motion Planning:** Outside of sampling-based motion planner, many other paradigms of motion planning make use of gradient information of some defined cost function. Trajectory optimisation approaches, such as CHOMP [14], STOMP[15] and TrajOpt [16], and potential field approaches [17] are prominent examples of this. Optimisation-based approaches are also central in controlling robots, to generate trajectories quickly, such as in crowded environments [18]. These approaches find a single solution by descending in the direction of lower cost, guided by the negative gradient. Likewise, our approach transform a distribution such that samples have lower cost by descending based on negative gradients.

**Diffeomorphisms and Normalising Flows:** The transforming of distributions via invertible and differentiable mappings is known as *normalising flows* [19]. These invertible and differentiable mappings are known as "flows" [19] or more generally as "diffeomorphisms" [20, 21, 22]. Normalising flows are typically learned using invertible structures [23, 24], with data drawn from the desired target distribution. We take a different approach, and develop invertible mappings from cost gradients, which can be hand specified or learned from other sources of data, such as occupancy information.

## 3    Parallelised Diffeomorphic Sampling-based Motion Planning

We shall detail the proposed Parallelised Diffeomorphic Sampling-based Motion Planning (PDMP) framework. In section 3.1, we begin by introducing neural network representations of occupancy,

which allows for fast batched querying of coordinates via the GPU. This occupancy representation will be used to construct diffeomorphisms which transforms a sampling distribution, such that samples have lower likelihood of being in occupied space. Details on constructing this diffeomorphism, along with those from an arbitrary cost function, are elaborated in section 3.2. Finally, in section 3.3, we expand on how we can leverage both the GPU, for the highly parallelisable transformation of samples, and the CPU, for the inherently sequential tree-building process, to achieve improved performance with the same time budget, for tree-building sampling-based motion planning methods.

## 3.1 Learning Continuous Occupancy Representations with Neural Networks

Occupancy in an environment have traditionally been represented by occupancy grid maps, which discretise the world into grid-cells and compute occupancy independently for each cell. Recent advancements in machine learning have brought continuous analogues of occupancy maps [25, 26], and continuous distance-based methods [27]. Here we present a straightforward approach of learning the occupancy via a neural network, which is fast to query, fast to obtain derivatives with respect to inputs, and inherently parallelised. These properties are beneficial for querying of large batches of coordinates.

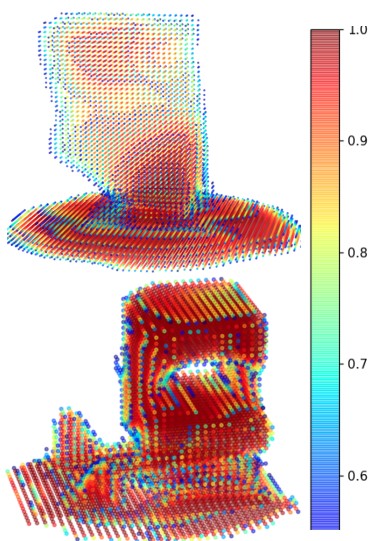

We are assumed to have a dataset of $n$ pairs, containing coordinates and a binary value, which indicates whether the coordinate is occupied, i.e. $\mathcal{D} = \{(\mathbf{x}_i, y_i)\}_{i=1}^{n}$, where $y_i \in \{0, 1\}$ for $i = 1, \ldots, n$. A dataset of this format can be obtained from depth sensors. Our aim is now to learn a mapping $f_{map}$ between a coordinate of interest $\mathbf{x}$ and the probability of being occupied at $\mathbf{x}$, $f_{map}(\mathbf{x}) = p(\mathbf{y} = 1|\mathbf{x})$. We shall model $f_{map}$ as a fully-connected neural network, with $tanh$ activation functions between hidden layers, and a $sigmoid$ activation layer at the output. The resulting setup is a binary classification problem, which can be learned via a binary cross entropy loss with gradient descent optimisers. Derivatives of the neural network can be obtained efficiently, via batched computation on a GPU.

Figure 1: Examples of continuous occupancy representation learned by a neural network (corresponds to Fig. 6).

## 3.2 Cost-informed Diffeomorphisms for Sampling Distributions

In this section we elaborate on building differentiable bijections, or diffeomorphisms, to transform a base distribution such that the "morphed" target distribution density is concentrated at where a provided cost function is low. That is, samples from the target distribution are more likely than the base distribution to be sampled from regions with low cost. Diffeomorphisms ensure that the transformed sampling distribution will have the same topology as the base distribution. For example, if the base distribution has infinite support, then the transformed sampling distribution also has infinite support, and will not have "holes" where there is no probability density.

**Constructing Diffeomorphisms via integral curves:** Diffeomorphisms can be generated by taking integral curves on the vector field defined by the negative gradients of the cost function. We consider an $n$-dimensional state vector $\mathbf{y} \in \mathbb{R}^n$ to be an initial time, provided a cost $f_c : \mathbb{R}^n \to \mathbb{R}$, an integral curve for some time $t \in \mathbb{R}$, can be written as an initial value problem (IVP):

$$\phi(\mathbf{y}) := \mathbf{y} - \int_0^t \nabla_{\mathbf{y}(s)} f_c(\mathbf{y}(s)) \mathrm{d}s = \mathbf{z}, \qquad \mathbf{y}(0) = \mathbf{y}, \qquad (1)$$

where $\mathbf{z} \in \mathbb{R}^n$ results from the Picard–Lindelöf theorem [28] (existence and uniqueness of IVPs), which states that if $\nabla_{\mathbf{y}(s)} f_c$ is Lipschitz continuous with respect to $\mathbf{y}(s)$, then the solution of the IVP exists and is unique. We shall restrict our discussion to cost functions with Lipschitz derivatives, this includes the continuous occupancy representations introduced in section 3.1. Then, $\phi(\mathbf{y})$ is a

diffeomorphism, and the inverse is given by:

$$\phi^{-1}(\mathbf{z}) := \mathbf{z} + \int_0^t \nabla_{\mathbf{z}(s)} f_c(\mathbf{z}(s)) \mathrm{d}\mathbf{s} = \mathbf{y}, \qquad\qquad \mathbf{z}(0) = \mathbf{z}. \qquad (2)$$

Therefore, we can use numerical integration techniques, such as Euler's method, to solve the IVP to evaluate the diffeomorphism efficiently.

**Bringing Diffeomorphisms into Configuration Space:** Motion-planning in robotics typically requires plans to be made in the *configuration space* (C-space) of the robot. On the other hand, costs to shape robot behaviour can be, and is often, defined in the Cartesian task space. For example, collision checking requires information about the task space geometry of the robot to determine whether it overlaps with objects in the environment. We assume that the sampling distribution is defined in the C-space of the robot, and diffeomorphisms need to operate in the C-space. We shall in particular discuss robot manipulators, where the states in the C-space are joint angles. We denote the C-space as $\mathcal{Q} \subseteq \mathbb{R}^n$, where there are $n$ joints. Joint configurations, $\mathbf{q} \in \mathcal{Q}$, are elements of the C-space, while Cartesian coordinates in task space are denoted as $\mathbf{x} \in \mathbb{R}^3$. We outline how to *pull* a cost gradient defined in the task space to the C-space, and construct a diffeomorphism there.

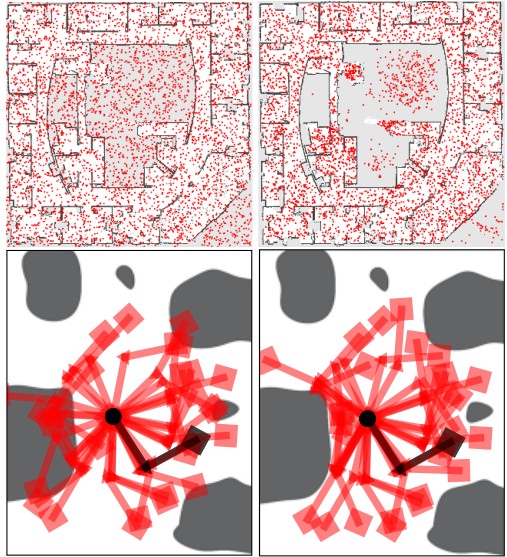

Figure 2: Examples of before and after diffeomorphism: (top) point robot; (bottom) 2 dof arm.

We start by defining $b$ body points on the robot, each with a forward kinematics function mapping configurations to the Cartesian coordinates at the body point, $\psi_i : \mathcal{Q} \to \mathbb{R}^3$, for each $i = 1, \ldots, b$. This allows us to make use of the Jacobian of the forward kinematics functions with respect to the joint configurations. The Jacobian of the $i^{th}$ kinematics function is denoted as $J_\psi^i(\cdot) = \frac{\mathrm{d}\psi_i}{\mathrm{d}\mathbf{q}}(\cdot)$. A cost potential $f_c$ which operates on the body points, such as the occupancy cost potential, can be *pulled* into the C-space:

$$\nabla_{\mathbf{q}} f_c = \sum_{i=1}^b J_\psi^i(\mathbf{q}) \nabla_{\mathbf{x}} f_c, \qquad (3)$$

we can then define a diffeomorphism, via solving the IVP as in (1), in the C-space of the robot. Fig. 2 illustrates two instances of diffeomorphism, from a (left) base to a (right) morphed distribution.

**Drawing Samples from the Morphed Target Distribution:** We can draw samples from the morphed distribution by drawing samples from the known base distribution, then passing the points through the diffeomorphism $\phi$. Unlike normalising flows for density estimation, which are computationally burdened by having to compute the determinant of the Jacobian of $\phi$, we only require the mapping of sampled points, which can be done efficiently, and does not require the Jacobian of $\phi$. Furthermore, we note that morphing the sampled points from the base distribution to the transformed distribution can be done in parallel if the cost gradients can be parallelised. In particular, occupancy gradients as introduced in section 3.1 can be batch computed on a GPU efficiently. In the following sections, we shall elaborate on how to exploit the parallel nature of the morphing sampled points.

### 3.3 Parallelised Diffeomorphic Transform of Sampling Distribution

Rapidly-exploring random trees (RRTs), its variants, along with its graph-based counter parts are some of the most widely-used motion planning algorithms. In this section, we develop the Parallelised Diffeomorphic Sampling-based Motion Planning (PDMP) algorithm to transform the sampling distribution while constructing trees, efficiently integrating CPU and GPU parallelism. We elaborate on the *diffeomorphic sampler*, which can be largely parallelised, and the *motion planner main thread*, which consists of sequential operations. An overview flow-diagram is shown in Fig. 3.

**Motion Planner Main Thread:** Building random trees is inherently a sequential process—sampling a random configuration, searching for nearest neighbour in the k-d tree, collision-checking of potential tree edges, connecting the candidate node , and rewiring of other existing nodes. This process requires knowledge of nodes that are currently connected by the tree, and valid nodes are connected to the tree as soon as they are found. Such a sequential process is repeated until the time budget exhausted. We shall denote this sequential process as the *Motion Planner* main thread. Similar to existing methods, we conduct the tree-building on the CPU. However, we proceed with optimising the sampled points in background threads that are parallelised in GPUs.

When the planning request is first received, the *main thread* spawns a background *boostrap thread* that prepares all necessary house keeping works such as constructing a concurrent queue $S$. Then, the *main thread* will proceed with the rest of the typical tree-building procedures, following the traditional RRT-variant literature. The main modification in this sequenital process lies in the sampling step. Typically, RRT samples from some distribution (e.g. uniform distribution $q_{\mathrm{rand}} \sim \mathcal{U}(0,1)$) within the same thread. Instead, in our PDMP framework we draw samples from the previously constructed concurrent queue $S$, which is one of the only communication contacts in between the *main thread* and the *diffeomorphic sampler thread*, to avoid any other synchronisation overhead (the other communication happens when the *main thread* requests the background *diffeomorphic sampler* to exits due to time budget being exhausted). This concurrent bucket is filled by our *diffeomorphic sampler thread*. In the event the planner attempts to draw from an empty bucket, sample points are immediately drawn from a simple prior distribution $\mathbf{q} \sim \mathcal{Q}_{prior}$, reverting back to a standard sampling-based planner. Therefore, the main thread does not need to do a blocking-wait on the background thread; which implies that, in the rare event of degraded GPUs performance, PDMP will only be reverting back to the typical planner performance.

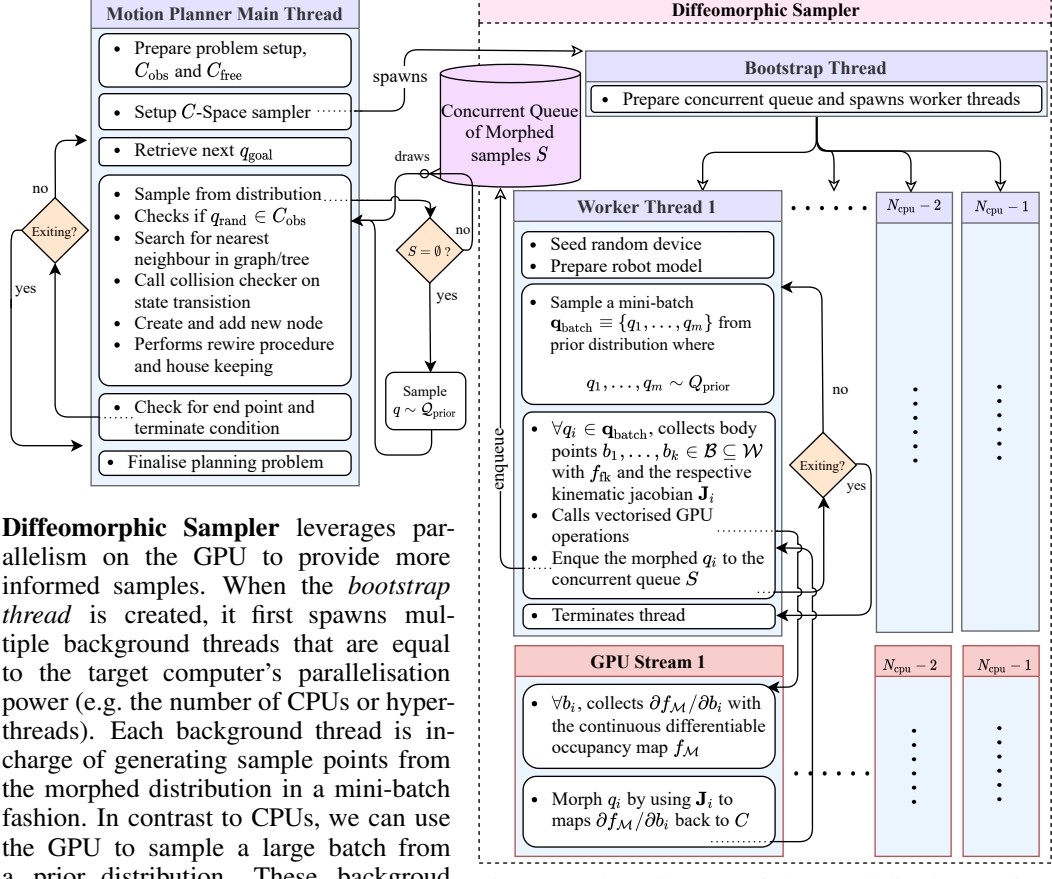

**Diffeomorphic Sampler** leverages parallelism on the GPU to provide more informed samples. When the *bootstrap thread* is created, it first spawns multiple background threads that are equal to the target computer's parallelisation power (e.g. the number of CPUs or hyperthreads). Each background thread is incharge of generating sample points from the morphed distribution in a mini-batch fashion. In contrast to CPUs, we can use the GPU to sample a large batch from a prior distribution. These background threads within *diffeomorphic sampler* in Fig. 3 will also collect the necessary

Figure 3: Flow diagram of the parallelised operations in PDMP.

kinematic Jacobians into batches for forward pass in the GPUs. The batch of samples is then passed through the diffeomorphism to obtain the informed samples. The pass through the diffeomorphism

can be done efficiently when leveraging the parallel computing capabilities of the GPUs, if the gradients of the cost potential can be done in batch. This is often the case if the cost gradients can be expressed analytically. This is particularly the case, if given by the derivative of a neural network.

### 3.4 Probabilistic Completeness

A particular benefit of morphing the sampling distribution, by a diffeomorphism, is that topology of the domain is preserved [29]. Intuitively, this means that if there are no "holes" in the original sampling distribution, there shall be no "holes" in the morphed distribution. The support of a probability distribution refers to the set of possible values of a random variable having non-zeros probability density. If the original sampling distribution is defined over an infinite support (such as if the base distribution is Gaussian), the transformed distribution also retains an infinite support, without any "holes". As such, drawing sample points from this transformed distribution shall maintains the probabilistic-completeness of RRT-based sampling-based methods [30].

In practice, the original sampling distribution is often defined on a finite support. Although there will be no holes in the transformed distribution, its support may be shifted. Additionally, parts of the space, where the model predicts to be occupied, can be stretched arbitrarily thin although still retaining non-zero probability density. Therefore, to maintain probabilistic completeness, one can deploy a strategy such as sampling with epsilon-bias towards a uniform distribution, as done in many learning-based sampling motion planning methods [10].

## 4 Experimental Results

We empirically analyse our proposed Parallel Diffeomorphic Sampling-based Motion Planning (PDMP) method. In the following sections, we investigate the performance of finding valid motion plans, with gradients from a cost potential occupancy representations, under time constraints. For our simulated environments, we construct three challenging environments, as illustrated in Fig. 5. *Divider*: consists of a large divider on a cluttered table, planning to reach the other sides of the divider; *Cupboard*: consists of a cupboard where the arm move in-between different shelves; *Lab-setup*: where the arm pickup an object and place it at the bottom of a cluttered scene.

### 4.1 Qualitative Evaluation on Informed Distribution

We hypothesise that after morphing our sampling distribution with a cost potential from our continuous occupancy representation, we can significantly improve the performance of the sampling-based planning strategies. In table 1 we evaluate the effect of PDMP on the sampling distribution, both quantitatively and in terms of feasibility. The results are broken down into samples that are contributed by the original uninformed prior distribution (top half) and by the informed diffeomorphic distribution (bottom half). The are no relative differences between the original and PDMP distribution

Table 1: Numerical results on (i) the total number of *samples* and (ii) percentage of *feasibility* on the original (Ori.) and PDMP distributions. The table illustrates that PDMP produces more feasible samples in all 3 environments. In the PDMP section, the *(from Prior)* and *(from Morphed)* breaks down the **Total** into samples that are from prior and morphed respectively. ($\mu \pm \sigma$)

|  |  | Environment | | |
|---|---|---|---|---|
|  |  | Divider | Cupboard | Lab-setup |
| Ori. | **Total** samples | 9543 ± 907 | 22606 ± 1375 | 13153 ± 554 |
|  | **Total** feasible | 52.19 ± 0.41 % | 12.77 ± 1.12 % | 48.38 ± 0.86 % |
| PDMP | **Total** samples | 10554 ± 751 | 23536 ± 1067 | 12985 ± 716 |
|  | *(from Prior)* | *17 ± 11* | *5322 ± 1278* | *294 ± 120* |
|  | *(from Morphed)* | *10536 ± 748* | *18214 ± 1190* | *12692 ± 736* |
|  | **Total** feasible | **81.52 ± 1.22 %** | **31.96 ± 2.23 %** | **72.83 ± 2.10 %** |
|  | *(from Prior)* | *51.02 ± 0.85 %* | *12.98 ± 1.92 %* | *49.12 ± 0.51 %* |
|  | *(from Morphed)* | *85.17 ± 1.74 %* | *37.51 ± 2.09 %* | *73.35 ± 1.82 %* |

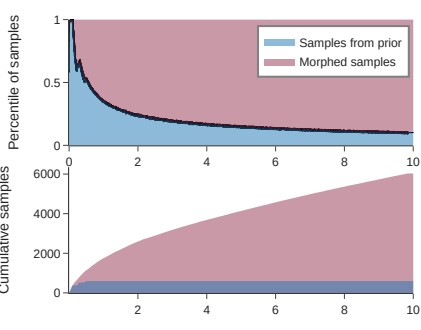

Figure 4: Percentage and cumulative counts of sample points drawn from an *uninformed prior*, and from a *transformed distribution*.

Table 2: Numerical results on various sampling-based motion planners on each environment. The *time-to-solution* refers to the time it took to obtain a solution trajectory (in seconds); and the *success pct.* refers to the percentage of runs that had successfully found a solution. Each SBP has a corresponding *PDMP* variant. Results are over 30 runs and with a time budget of 20 seconds. Results shown are mean $\pm$ one standard deviation ($\mu \pm \sigma$).

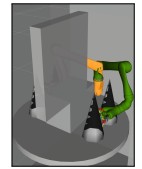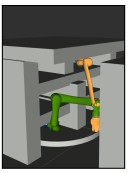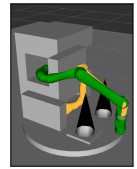

Figure 5: Left to right are environments of *divider*, *cupboard* and *lab-setup*.

| | Planner | Environment | | |
|---|---|---|---|---|
| | | Divider | Cupboard | Lab-setup |
| **Time-to-solution** | RRT* | 13.83 ± 7.72 | 18.79 ± 4.53 | 12.56 ± 7.78 |
| | PDMP-RRT* | 5.99 ± 3.98 | 17.01 ± 6.14 | 9.26 ± 8.37 |
| | RRT.C* | 2.72 ± 0.91 | 5.51 ± 7.38 | 1.60 ± 0.63 |
| | PDMP-RRT.C* | **1.60 ± 1.26** | **3.64 ± 5.76** | **1.29 ± 0.44** |
| | L.PRM* | 14.98 ± 7.96 | N/A | 16.57 ± 6.93 |
| | PDMP-L.PRM* | 13.37 ± 7.44 | N/A | 14.71 ± 7.07 |
| | STOMP | 14.83 ± 4.94 | 14.54 ± 5.90 | 14.29 ± 6.32 |
| **Success pct.** | RRT* | 43.33 % | 6.67 % | 53.33 % |
| | PDMP-RRT* | 96.67 % | 20.00 % | 70.00 % |
| | RRT.C* | **100 %** | 83.33 % | **100 %** |
| | PDMP-RRT.C* | **100 %** | **93.33 %** | **100 %** |
| | L.PRM* | 33.33 % | 0 % | 20.00 % |
| | PDMP-L.PRM* | 43.33 % | 0 % | 46.67 % |
| | STOMP | 63.33 % | 53.33 % | 53.33 % |

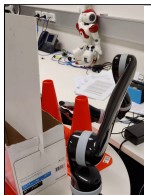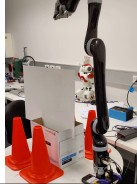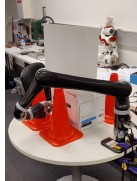
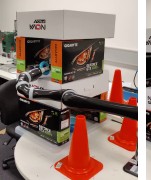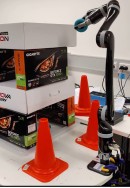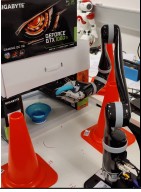

Figure 6: Sequence of trajectory in the real-world experiments with the Jaco arm: (Top) *Divider*; (Bottom) *Lab-setup*.

in *total samples*, this means that PDMP does not slow down the drawing of random samples. Instead, the morphed samples are more beneficial to the planning problem as they are more likely to be feasible in free space, as shown by the higher *total feasible* percentage in the PDMP section.

## 4.2 Higher Success Rates with Informed Sampling Distributions

We examine our hypothesis by testing various sampling-based motion planners (SBPs) within our PDMP framework. We investigate three SBPs—RRT* [31], RRT*-connect [32], Lazy-PRM* [33], and a trajectory optimisation-based planner STOMP [15]. For the SBPs, we compare the effect of morphing the sampling distribution provided by our approach to standard uninformed sampling.

We provide a time-budget of 20 seconds for each planner, and calculate the percentage of tries, over 30 runs, that result in a successful plan at different times until the budget was entirely used. The morphing of distribution is implemented under [34]. The results are illustrated in Fig. 7. We see that for each of the three sampling-based planning methods, incorporation within the PDMP framework to morph the sampling distribution improves the success rate. This is most evident when using RRTs within PDMP, since it allows us to produce more successful samples which in-turn speed up the tree-building process. We replicate the *divider* and *Lab-setup* environment (Fig. 6) in the real-world with a 6-DOF Jaco manipulator. The planning is illustrated by videos included in the supplementary materials.

Table 2 illustrates numerical results of two important attributes in motion planning—the *time-to-solution* and the *success percentage*. Overall, PDMP allows each motion planners to utilise sampled configurations that are more likely to be feasible (see table 1), which in turn allow PDMP planners to achieve shorter *time-to-solution* when compared to their original counterpart in table 2. Therefore, they are also more likely to successfully obtain a solution trajectory within the allocated time budget.

In the *divider* environment, we see that the success rate of PDMP-RRT* reaches 80% at around 7 seconds of planning, and almost has a perfect success rate by the end of the 20s budget. On the other hand, vanilla RRT* with a uniform sampling distribution has a success rate of under 50% when the time budget is used up. The same trends are observed with the other variants. Overall, the RRT*-connect within PDMP and with a uniform sampling distribution outperforms the other variants. Even still, when using RRT*-connect within the PDMP, we observe higher success rates when the planning time is low (under 3 seconds), indicating that PDMP significantly improves time-to-solution. The

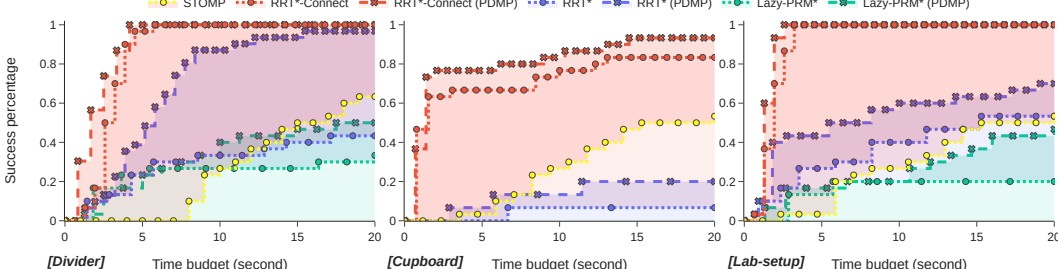

Figure 7: The success rate of the planning algorithm variants, over 30 runs. We observe that PDMP enables all flavours of sampling-based motion planning algorithms to have improved success rates, particularly at lower planning times.

time-to-solution of STOMP tends to spread out among all three environments, which is likely due to its stochastic nature. The performance of STOMP does not seem to be degraded by the complexity of the environment, which suggests that the cost information was able to guide STOMP to obtain a solution trajectory. Our PDMP framework provides clear imporvements to the success rates of both RRT* and RRT*-connect methods. Additionally, we observe that at the end of the 20s time budget, PDMP variants outperform their counterparts which sample from an uninformed sampling distribution. Lastly, Lazy-PRM* performs poorly on most environments as it is a multi-query planner, and was not able to obtain any valid solution in *Cupboard* within the allocated time budget.

## 4.3 Influence of CPU-GPU Parallelisation

Our PDMP method parallelises over the CPU and GPU, by allocating the GPU to filling up a bucket for which samples are drawn, and dedicating the CPU to the planning process, which typically involves building a tree. If the bucket is empty when a sample is needed for the sequential planning process on the CPU, a sample is drawn from an uninformed prior distribution. Therefore, obtaining informed samples come at almost no additional cost: in the worst-case scenario, if the planning process faster than drawing samples from the morphed distribution, PDMP falls back to a vanilla sampling based algorithm, drawing samples from a uninformed distribution.

Intuitively, the more samples obtained from the bucket used in the planning process, the more informed our used samples are. We investigate the number of sample points drawn from the bucket, which are from the "morphed" distribution, and the percentage of samples from the uninformed prior as planning time progresses. This is shown in Fig. 4. We observe at the beginning, as the bucket has not yet been filled with samples from the morphed sampling distribution, samples from the uninformed prior are used. However, the GPU is able to quickly fill up the bucket with samples from the morphed distribution, and the number of uninformed samples beyond 0.2s is largely negligible. By 1 second of sampling time, the cumulative "morphed" sample points significantly exceeds the cumulative uninformed samples. This indicates that at any reasonable amount of planning time, the process of drawing samples from an informed distribution is much faster than the main planning process.

## 5 Conclusions

In this paper a novel method combining cost gradients from optimisation-based motion planning with probabilistic-complete sampling-base motion planning methods is proposed. Parallelised Diffeomorphic Sampling-based Motion Planning is a motion-planning framework which utilises *diffeomorphisms* generated from gradients of defined cost to morph the sampling distribution for sampling-based motion-planning methods. We demonstrate how such diffeomorphisms can be created from learned models of environment occupancy to encode obstacle avoidance behaviour, or user specified biases. Additionally, an implementation which parallelises this process across the GPU and CPU is provided, showing that sampling from the more informed distribution can be achieved at no additional run-time cost. We empirically demonstrate that our method is capable of significantly improving the success rate of finding solutions in challenging planning environments.

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
