# OpenReview forum: "Parallelised Diffeomorphic Sampling-based Motion Planning"
_robot-learning.org/CoRL/2021/Conference — CoRL2021 Poster_

### Official Review · Reviewer_9VQ2 · 2021-07-19

**Originality:** Excellent
**Technical Quality:** Excellent
**Clarity Of Presentation:** Excellent
**Impact:** 4

**Recommendation:**

Strong Accept: I recommend accepting the paper and will argue for my recommendation even if other reviewers hold a different opinion.

**Summary:**

The paper presents Parallelised Diffeomorphic Sampling-based Motion Planning (PDMP), a novel parallelised framework which leverages diffeomorphisms to transform sampling distributions of sampling-based motion planners based on specified costs (such as those for obstacle avoidance) in the more favorable space (Cartesian space) rather than in the Configuration Space. The paper presents the algorithm which in the baseline scenario will revert back to the performance of the original Sampling-based Motion Planning (SBMP) algorithm (such as the RRT, RRT-connect, etc), but in general improves upon the performance of these algorithm in terms of higher success rate and lower planning times. In the experimental section, the PDMP algorithm is benchmarked against the original SBMP algorithms, as well as tested for motion planning on a real robot setup. Details are also provided for parallelizing the algorithm with GPU computations.

**Issues:**

While Equations (1), (2), and (3) explains how diffeomorphisms are leveraged to transform the cost vector field from the task/Cartesian space to the Configuration (C) space, I am not very clear on how the sampling distributions are transformed. Can the authors please provide some concise equations that explain the sampling distribution transformation?

**Reviewer Expertise:**

Very good: Comprehensive knowledge of the area

**Strengths And Weaknesses:**

Strengths:
(1) The paper is well-written. To be honest: I really enjoyed reading the paper! Great work!
(2) Implementation code is attached in the Supplementary Materials.

Weaknesses:
Only some minor things as mentioned in the `Issues` section.

**Summary Of Recommendation:**

This is an excellent paper. I strongly recommend to accept the paper.

---

> ### Author Response · Authors · 2021-08-26
> **Addressed question on transforming distribution in C-space**
>
> We thank the reviewer for their feedback and appreciate their comments highlighting the novelty and impact of our submission. The suggestions in the improvements section are helpful to make the paper more accessible and we have incorporated them in the new version. We also want to thank the reviewer for the positive feedback and pointing out the novelty of this paper. For the issues/necessary clarification that the reviewer had pointed out, they are elaborated as follows:
>
>
> > While Equations (1), (2), and (3) explains how diffeomorphisms are leveraged to transform the cost vector field from the task/Cartesian space to the Configuration (C) space, I am not very clear on how the sampling distributions are transformed. Can the authors please provide some concise equations that explain the sampling distribution transformation
>
>
> As the reviewer had rightly pointed out, we had utilised equations (1) and (2) with numerical integration to map the cost functions, via integral curve, back into the Configuration Space via (3). Similarly, we can utilise the same concept to morph the sampled base distribution into a morphed distribution (e.g. Fig 2).
>
> We begin by empirically drawing a set of configurations $\mathbf{q}_\text{batch} \equiv \\{ q_1,\ldots,q_n \\}$ from a prior distribution $q_1,\ldots,q_n\sim\mathcal{Q}_\text{prior}\subseteq\mathbb{R}^n$. Then, for each drawn configuration from the batch $q_i \in \mathbf{q}_\text{batch}$, we collect multiple body points $x_1,\ldots,x_b\in\mathcal{B}$ on the robot arm via forward kinematic $\psi_i$, where $\mathcal{B}\subseteq\mathbb{R}^3$ denotes the space of all body points. Note that at the same time we also collect the Jacobian of the forward kinematic functions here. Now that we have obtained a set of body points in $\mathbb{R}^3$, we will collect the corresponding gradient of cost information from our learned continuous occupancy map in $\mathbb{R}^3$. We can then map the gradient information from our $b$ body points from $\mathbb{R}^3$ back into C-space via the corresponding Jacobian of the forward kinematics functions _wrt_ the joint configuration $q_i$ (the ones that we had collected previously). Hence, the gradient information from body points is mapped back into C-space. The procedure Is illustrated in Fig 3.
>
> Mathamatically, the mapping of gradient information from worldspace $\nabla_{\mathbf{x}}$ to C-space $\nabla_{q}$ is expressed as eq 3:
> \begin{equation}
>     \nabla_{\mathbf{q}}f_{c}=\sum_{i=1}^{b}J_{\psi}^{i}(\mathbf{q})\nabla_{\mathbf{x}}f_{c},
> \end{equation}
> with $J_{\psi}^{i}(\cdot)=\frac{\mathrm{d}\psi_{i}}{\mathrm{d}{q}}(\cdot)$ denoting the $i^{th}$ kinematics function and $f_{c}$ denoting cost potential that operates on the body points. Each previously drawn configuration in $\mathbf{q}_\text{batch}$ will then follows its gradient information $\nabla_\mathbf{q}$ and maps to the morphed space.

---

### Official Review · Reviewer_QAiv · 2021-07-19

**Originality:** Fair
**Technical Quality:** Good
**Clarity Of Presentation:** Fair
**Impact:** 2

**Recommendation:**

Weak Reject: I recommend rejecting the paper, but will not argue for my recommendation if the majority of other reviewers have a different opinion.

**Summary:**

The paper is about a new motion planning framework called PDMP (Parallelised Diffeomorphic Sampling-based motion Planning), which morphs the sampling distribution in the state space of the robot using external cost functions. This framework can be combined with any motion planner and basically replaces the sampling step of each planner with a new informed sampling framework. This informed sampling framework works by employing a diffeomorphism, which maps samples towards a lower cost function in the state space. This creates more valuable samples, which have a higher probability of being collision-free. In two experiments, the authors demonstrate that their framework can be applied to two manipulation scenarios and that any motion planner can be substituted into their framework.

**Issues:**

-- Please add a section where you discuss completeness (and optimality) guarantees (see comment above)

-- Remove Sec. 4.3 or add a better motivated cost function

-- Add more experiments and/or domain randomization to strengthen the case for your framework. Please also include Lazy-PRM in the cupboard experiment (or remove it from the divider)

**Reviewer Expertise:**

Very good: Comprehensive knowledge of the area

**Strengths And Weaknesses:**

Strengths:

-- Having a good sampling distribution is key to plan efficiently through high-dimensional state spaces. I fully agree with the authors that it makes sense to think more deeply about how such a sampling distribution could look like and how we can deform/map samples to make them more valuable.

Weaknesses:

-- Your approach works by sampling uniformly, then mapping those samples towards a lower cost in the state space. While I think this idea make sense, it is unclear if you can actually keep completeness (and optimality) guarantees. You use also uniform sampling if there are no informed samples, however, as your Fig. 3 demonstrates, the number of uniform sampled points might (in the limit) converge to zero. In that case, I don't think you could still argue for completeness. I would therefore recommend that the authors think more deeply about how to keep completeness in this framework or by adding a small epsilon bias of uniform sampling. In any case, a discussion of those properties seem to be more valuable compared to the current discussion in Sec. 4.3 for example (see comment below).

-- In Sec. 4.3 you introduce an additional cost which adds a higher probability of getting samples close to zero radians. However, it is unclear why you would do that. There is no motivation given in this section, which would let me see how a zero radian sample might be better than a random sample. It seems the purpose of this whole section is just to demonstrate that combined cost functions are possible. However, I would either remove this section or better motivate why a zero radian cost makes sense in a manipulation task.

-- In Sec. 4.1 you argue that PDMP improves the finding of solutions significantly. However, you (1) did not say what you mean by significantly, and (2) just demonstrating the algorithm on two scenarios is not sufficient to make such claims. The biases you introduced could also be implict biases for those specific tasks you evaluated on. It would be better if you could add more variations or more scenarios to strengthen your claim that PDMP improves upon the given sampling algorithms.

Minor Comments:

-- Consider using a different colorscheme for the robots in Fig. 5, they are difficult to distinguish in b/w print.

-- Fig. 3 should come before Fig. 4, not after Fig. 6.

-- You did not label the axes in Fig. 6

-- Missing Lazy-PRM in Fig. 4 (right). Also: which experiments are shown here? I guess it is the divider and the cupboard. Please clarify.

-- Line 322: "from defined from" -> "from" ?
-- Line 153: "the states its C-space" -> "the C-space" ?
-- Line 155: gradients -> gradient
-- Line 168: "the the Jacobian" -> "the Jacobian"
-- Line 123: "is a" -> "as a"
-- Line 140: "is the resulting By the" -> Unclear. Missing sentence part!?

**Summary Of Recommendation:**

While the paper shows promise, the authors have only evaluated their algorithm on two scenarios without randomization, which makes it rather difficult to assess if their method is really beneficial. Also, there is no discussion about completeness/optimality, which I would have expected in a more planning-oriented paper. There were also several instances of sloppy writing and unlabeled axes, which further shows that the paper is still a bit immature and needs further improvement.

---

> ### Author Response · Authors · 2021-08-26
> **Part 1: Addressed concerns on completeness/optimality guarentees, section on additional cost, various figures and typos**
>
> We thank the reviewer for their feedback and appreciate their comments highlighting the novelty and impact of our submission. The suggestions in the improvements section are helpful to make the paper more accessible and we have incorporated them in the new version.
>
> > it is unclear if you can actually keep completeness (and optimality) guarantees...... I would therefore recommend that the authors think more deeply about how to keep completeness in this framework or by adding a small epsilon bias of uniform sampling.
>
> Thank you for the excellent suggestion. We had theoretically analysed our method and it is presented in section 3.4 now. To summarise, the probabilistic completeness guarantee is still held when the prior and the transformed probability distribution are equal. This is given by the bijective property of diffeomorphism. We had also discussed the use of epsilon bias in the text.
>
> > In Sec. 4.3 you introduce an additional cost which adds a higher probability of getting samples close to zero radians...... There is no motivation given in this section... It seems the purpose of this whole section is just to demonstrate that combined cost functions are possible.
>
> Thank you for pointing that out. You are correct in saying that the main purpose of the whole section is for us to demonstrate that our method allows other arbitrary cost functions (even though we had focused on the collision cost in the paper as it's the most common one). The rationale behind the choice of zero radians (assuming the joints rotate from -pi to pi) is that there are scenarios where it is more beneficial to bias closer to the center of the rotational joint, such that the arm will have more freedom to move with more degree of freedom later on in the path.
>
> > a discussion of those [completeness] properties seem to be more valuable compared to the current discussion in Sec. 4.3 for example
>
> We agree with the reviewer and removed section 4.3 in favour of the discussion on probabilistic completeness in section 3.4 (because of space constraints).
>
>
> > In Sec. 4.1 you argue that PDMP improves the finding of solutions significantly. However, you (1) did not say what you mean by significantly, and (2) just demonstrating the algorithm on two scenarios is not sufficient to make such claims... It would be better if you could add more variations or more scenarios
>
> We apologise for the confusion. “...PDMP improves the findings of solutions significantly.” was trying to refer to the fact that it is easier for PDMP to find solutions (i.e. qualitative, it has a faster time-to-solution, see Table 2). We had to clarify that in the text. We had also included an additional scenario (named as “Lab-setup” in Fig. 4, 5). The Table 1 and 2 are also additional numerical results to strengthen the claim.
>
>
> > Consider using a different colorscheme for the robots in Fig. 5, they are difficult to distinguish in b/w print.
>
> Thank you for the suggestion, and we had switched the colorscheme of that figure (now it is Fig 5) from blue/green to green/bright orange, which has a stronger contrast in grayscale.
>
> > Fig. 3 should come before Fig. 4, not after Fig. 6.
>
> Thank you for pointing that out, it was some artifact from using wrapfig in latex. The figures are ordered in the new text.
>
> > You did not label the axes in Fig. 6
>
> Thank you for pointing that out, currently, Fig. 6 is removed along with sec. 4.3 as per your previous suggestion.
>
> > Missing Lazy-PRM in Fig. 4 (right). Also: which experiments are shown here? I guess it is the divider and the cupboard. Please clarify.
>
> We apologize for missing the label of the subfig in Fig. 4, the figure (now Fig 7) is now labelled with an additional subfig. For Lazy-PRM*, it was not missing but it is the fact that none of the Lazy-PRM* instances can complete the planning scenario within the time budget. It was indicated in the old text line 273-274. Perhaps it was not too clear, and we had further clarified that fact in the new text line 318.
>
> > -- Line….. [various typos]
>
> Thank you very much for pointing out a lot of typos and marking the line for each of the typos/mistakes that you had discovered. We really do appreciate your notes on those and we had fixed all of them. We had also further proofread the paper.
>
> ----------
>
> _>> [see next comment]_

---

> > ### Author Response · Authors · 2021-08-26
> > **Part 2: Addressed concerns on randomisation, probabilistic completeness and optimality**
> >
> > _>> [continued from previous comment]_
> >
> > ----------
> >
> >
> >
> > > only evaluated their algorithm on two scenarios without randomization, which makes it rather difficult to assess if their method is really beneficial.
> >
> > We want to clarify that we **did not fix any random seeds** and there exists randomisation within the planning scenarios (hence the “success percentage” in table 2 and the cumulative plot in fig 6). All sampling-based motion planners and trajectory optimisation-based planners contain randomness.
> >
> > In our investigation: for all 7 (original + variants) planners, each planner is executed on 3 environments, each environment is repeated 30 times for statistical significance. If the randomisation is referring to start and goal configuration randomisation, we want to point out that currently each scene has a semantic meaning of the planning trajectory and it does not make too much sense to randomly pick two points in free space (and most pairs of points are highly likely to have trivial solutions). Moreover, it is hard to perform statistical analysis across all pairs of start/goal pairs as each pair will have highly varying traj-length and time-to-solution, which would render results in table 2/fig 7 uninformative.
> >
> > > - Also, there is no discussion about completeness/optimality, which I would have expected in a more planning-oriented paper.
> > > - Please add a section where you discuss completeness (and optimality) guarantees (see comment above)
> >
> > Discussion on completeness is now in section 3.4. With regards to optimality: while we believe that we did not break any assumptions used by RRT*-like sampling-based planners for proving asymptotic optimality (AO) guarantee, as of now we have not written up rigorous formal proof due to time constraints. The main ingredients of proofs for most AO guarantee relies on the fact that (i) the planner is probabilistic complete (PC) and samples cover everywhere, (ii) adequate tree/graph building procedures where there is a non-zero probability that the construction of the tree/graph will have opportunities to consider all paths that lead to $q_i\in\mathcal{Q}$ (e.g. rewiring procedure in RRT* with shrinking connection radius). Since our framework remains to be  PC and our framework did not touch on the actual graph building procedure, we believe PDMP can directly inherent AO from the inner planner (e.g. RRT*, PRM*, etc.). However, the write up of a formal proof will need to wait for future works.

---

> > > ### Comment · Reviewer_QAiv · 2021-09-04
> > > **Response to rebuttal**
> > >
> > > I appreciate that the authors took the time to respond to my review. While I feel that many of my points have been addressed, I have some concerns about the proof of completeness. In particular, your whole proof depends on eq. 4 of the referred paper. However, the proof seems to be different from what you claim in the paper. It would be good if the authors could clarify the proof in future work, especially how the support actually is supposed to be equivalent. A simple example or intuitive explanation would be appreciated. In the current version, I have doubts about the validity of the proof. Furthermore, I still believe the number of experiments not to be enough to demonstrate the usefulness of the approach. I therefore keep my score as is.

---

### Official Review · Reviewer_mUoR · 2021-07-23

**Originality:** Very Good
**Technical Quality:** Fair
**Clarity Of Presentation:** Very Good
**Impact:** 4

**Recommendation:**

Weak Reject: I recommend rejecting the paper, but will not argue for my recommendation if the majority of other reviewers have a different opinion.

**Summary:**

Combines sampling-based planning with gradient-based information, as a way to both globally explore the space but also exploit local cost information. The idea is to start with a regular sampling distribution (called base distribution) and to morph it into a more informative obstacle-aware distribution. In practice, the paper actually does not morph the distribution, instead it takes individual samples and propagates them along the integral curves of the vector field defined by the cost gradients. This results in samples at some "earlier" time t of the vector field flow which should have a higher chance to be away from obstacles (assuming the cost encodes obstacle distance).

The authors describe an algorithm that relies on parallelization for propagation samples on the CPU for building the tree. Experiments with robotic manipulator reaching constrained targets are presented with comparison to standard RRT methods.



**Issues:**

- simple 2d environment with obstacles and some simple manipulator (e.g. even with 2dof is fine) , showing the base samples and how they are propagated will be useful
- explanation of why the occupancy cost gradients even make sense for this application

**Reviewer Expertise:**

Very good: Comprehensive knowledge of the area

**Strengths And Weaknesses:**

+ the paper presents a novel idea for adjusting samples along cost gradients to increase their quality
+ the method naturally exploits parallelism so could be done efficiently
+ a NN is used for the occupancy which gives a differentiable cost, the gradient of which is used for the flow

- it is not clear what happens if a sample from the base distribution is in the interior of a solid obstacle where basically the occupancy gradient will be zero, the paper doesn't explain how this is addressed. It almost feels like the method would work only for samples at the boundary of an obstacle?
- it is not clear for how long the samples need to propagated along the gradients
- it is not clear how the cost gradient is computed for an arbitrary environment (it's
- would have been great to see an application to simple planar obstacle environment as an illustration of the method, and to see how the deformed samples look like, there isn't a singe picture in the paper illustrating the sample deformation


Other minor comments:
- not clear what Figure 1 shows, is there a corresponding real-world scene that can be added for clarity?
- line 140 typo: "resulting By"

**Summary Of Recommendation:**

The idea of using gradients from differentiable NN cost and propagating samples along them is promising. Applying this to occupancy gradients though seems to a bit contrary to the original motivation. A simple intuitive and visual illustration of the method would have helped but is lacking.

---

> ### Author Response · Authors · 2021-08-26
> **Addresed concerns on zero gradients, propagation time, 2D visualisations, issues with figs and typos**
>
> We thank the reviewer for their feedback and appreciate their comments on the novelty, strengths, and recommendations on our submission. The suggestions in the weaknesses section are helpful to strengthen the paper, and we have addressed all of the issues as detailed in the following:
>
> > “it is not clear what happens if a sample from the base distribution is in the interior of a solid obstacle where basically the occupancy gradient will be zero, the paper doesn't explain how this is addressed”
>
> The gradients in the interior of the obstacles do diminish as we get away from the boundary. However, this is fairly gradual especially with regularisation applied, i.e. the occupancy across obstacle boundaries is modelled not as a sharp step. We emphasise that we bias our samples to be in free areas, and do not claim to draw samples only from only free space, that is we still need to verify the potential node. Empirically, we show the improvement brought by this biasing is sufficient to obtain great performances improvements. See Figure 2 as a visual example of the effect of the diffeomorphism on distributions.
>
> > “it is not clear for how long the samples need to propagated along the gradients”
>
> The propagation time is a hyperparameter that had been preset before the problem instance. We did not discuss the propagation time in the paper, however, it is a trivial task to tune a good propagation time by evaluating the diffeomorphism after finishing training the continuous map. We use the same propagation time among all environments. (it might be beneficial to use different propagation times in different local areas of the environment, however, we did not investigate that).
>
> > “it is not clear how the cost gradient is computed for an arbitrary environment”
>
> We first obtain 3D point cloud data that encodes the occupancy information of a new environment. Then, the point cloud is used to train a continuous occupancy representation of the environment (as described in section 3.1). The "cost gradient" would then be referring to the gradient of the predicted occupancy value by our continuous occupancy map.
>
> > “would have been great to see an application to simple planar obstacle environment as an illustration of the method, and to see how the deformed samples look like.
> > Simple 2d environment with obstacles and some simple manipulator …”
>
> We now include two 2D visualisations of the base and morphed distribution in Figure 2. One of the visualisations uses a point robot on a 2D map, and the other uses a 2 dof link arm on a 2D map.
>
> > “not clear what Figure 1 shows, is there a corresponding real-world scene that can be added for clarity?”
>
> We had included photos of the corresponding real-world scene (in Figure 6), and Figure 1's caption now refers the audience to the corresponding scenes.
>
> > “line 140 typo: "resulting By"”
>
> Thank you for pointing this out, and we have fixed the typo.
>
> > “explanation of why the occupancy cost gradients even make sense for this application”
>
> The reason that we had chosen occupancy value as the cost to minimise is due to the fact that we would want to maintain clearance between the manipulator and the obstacles. Additionally, following the occupancy gradients of continuous occupancy models ensures that integral curves are valid diffeomorphisms. This is to account for any sensor or controller execution noise that is common among hardware. It is also a common choice in trajectory optimisation methods, such as  [1].
>
>
> [1] Francis, Gilad, Lionel Ott, and Fabio Ramos. "Fast stochastic functional path planning in occupancy maps." 2019 International Conference on Robotics and Automation (ICRA). IEEE, 2019.

---

### Official Review · Reviewer_Dafq · 2021-07-24

**Originality:** Good
**Technical Quality:** Very Good
**Clarity Of Presentation:** Very Good
**Impact:** 3

**Recommendation:**

Weak Accept: I recommend accepting the paper, but will not argue for my recommendation if the majority of other reviewers have a different opinion.

**Summary:**

The paper proposes Parallelized Deiffeomorphic Sampling-based Motion Planning (PDMP) to combine the advantages of trajectory optimization and sampling-based methods for planning. The method borrows ideas from normalizing flows and leverages gradient information to speed up motion planning. A complete diagram of the flow is provided. The experiments are conducted based on two scenarios to show that the proposed method can have higher success rates.

**Issues:**

The major issues are in the experimental validation.

**Reviewer Expertise:**

Good: General knowledge of the area

**Strengths And Weaknesses:**

Strengths

- The attempt to combine trajectory optimization and sampling method for planning is interesting. It can potentially bring about some new ideas in this area.
- Experiments on real robots are conducted to verify the algorithm. It is also compared with some traditional sampling methods to show that they can have higher success rates.
- Parallelization is considered to accelerate the implementation of the methods. This optimization effort can make the method more practical.

Weaknesses

- The experiment validation is limited. Only two scenarios are considered in the evaluation.
- Also, no comparisons with the trajectory optimization method are made.

**Summary Of Recommendation:**

The overall idea and approach are interesting. Despite the concerns on experiment validation, I am leaning towards acceptance.

---

> ### Author Response · Authors · 2021-08-26
> **Addressed concerns on experiment (new environment) and comparisions (new trajectory optimisation-based planner)**
>
> We thank the reviewer for their feedback and appreciate their comments on the novelty, advantages, as well as limitations of our submission. The suggestions in the weaknesses section are helpful to strengthen the paper, and we address all of the issues as detailed in the following:
>
> > The experiment validation is limited. Only two scenarios are considered in the evaluation.
>
> [Experiment] We incorporate an additional environment (named as "Lab-setup" in Fig. 5, Fig. 6) both in simulation and as a real-world experiment. We also include more numerical results to strengthen the experimental sections (Table 1 and 2) which summarise the total number of feasible and infeasible sampled configurations, time-to-solution, and success percentage.
>
> > Also, no comparisons with the trajectory optimization method are made.
>
> [Comparisons] We include an additional planner, STOMP, as a widely-used and representative trajectory optimization-based motion planner in all experimental plots and all numerical results.

---

### Meta-Review · Area_Chair_JNMn · 2021-08-15

**Recommendation:** Accept (Poster)
**Confidence:** 4

**Metareview:**

In this work, the authors propose an approach to combine the respective advantages of the sampling-based and optimization-based planning methods. The reviewers agree regarding the motivation of the approach and the relevance to the field.

The major concerns are related to the experimental validation and the results -- validation of the proposed approach on additional systems and/or environments (with multiple different initial seeds), and comparisons with additional optimization-based baselines will significantly strengthen the paper.

The paper will benefit from a clear discussion on the completeness and optimality of the proposed approach and the addition of some visualizations on sampling distribution transformation (even if for simple planning problems).

Besides addressing the major issues mentioned above, the authors should also revise the manuscript according to the other clarifications requested and the suggestions provided.

===== Post rebuttal =====

The authors have sufficiently addressed reviewers' concerns; specifically, including additional experiments and baselines was particularly helpful. For the final version, it is still important to address the concerns of Reviewer QAiv regarding the completeness proof.

---

> ### Author Response · Authors · 2021-08-26
> **Listing of all changes in the new text**
>
> We thank the area chair for their feedback and appreciate their comments highlighting the novelty and impact of our submission. The suggestions in the improvements section are helpful to make the paper more accessible and have been incorporated in the new version. We have addressed every comment and recommendation by all four reviewers in the new text. In particular:
>
> 1. We have removed one figure (Fig. 6 in old text) and one section (sec 4.3 in old text) as per suggestions by reviewers. The updated text currently contains 4 new/updated figures (Fig 1, 2, 5, 6, 7) and 2 new tables. We have replaced sec 4.3 with sec 3.4 to facilitate discussion on probabilistic completeness.
>
> 2. **[Experiments]** For various concerns related to the experimental validation and results
>     1. We introduce an additional environment (named as *"Lab-setup"* in Fig. 5, Fig. 6) both in simulation and as a real-world experiment. The environment is included in the main plot Fig 7.
>     1. We include additional numerical results to strengthen the experimental sections (Table 1 and 2) which summarise the total number of feasible and infeasible sampled configurations, time-to-solution, and success percentage.
>
> 3. **[Comparisons]** For concerns regarding the comparisons:
>     - We include an additional planner, STOMP, as a widely-used and representative trajectory optimization-based motion planner in all experimental plots and all relevant numerical results.
>
> 4. **[Completeness]** For concerns on completeness:
>     - We theoretically analyse our method and it is now presented in section 3.4.
>
> 5. **[Visualisation]** For concerns on additional visualisation:
>     1. We include two 2D visualisations of the base and morphed distribution in Figure 2.
>     1. The real-world scenarios is now a sequence of configurations that better visualise the trajectory
>
> 6. We had also addressed all concerns and questions raised by the reviewers, as detailed in their separate comments below.

---

### Decision · Program_Chairs · 2021-09-13

**Decision:**

Accept (Poster)

**Comment:**

In this work, the authors propose an approach to combine the respective advantages of the sampling-based and optimization-based planning methods. The reviewers agree regarding the motivation of the approach and the relevance to the field.

The major concerns are related to the experimental validation and the results -- validation of the proposed approach on additional systems and/or environments (with multiple different initial seeds), and comparisons with additional optimization-based baselines will significantly strengthen the paper.

The paper will benefit from a clear discussion on the completeness and optimality of the proposed approach and the addition of some visualizations on sampling distribution transformation (even if for simple planning problems).

Besides addressing the major issues mentioned above, the authors should also revise the manuscript according to the other clarifications requested and the suggestions provided.

===== Post rebuttal =====

The authors have sufficiently addressed reviewers' concerns; specifically, including additional experiments and baselines was particularly helpful. For the final version, it is still important to address the concerns of Reviewer QAiv regarding the completeness proof.